# QuRL: Rubrics as Judge For Open-Ended Question Answering

**Xiyu Wei**[1,2*]**, Qingwei Zong**[1,3*]**, Xiaoguang Li**[4]**, Eugene J. Yu**[1,3]**, Sujian Li**[1,3†]

[1] Key Laboratory of Computational Linguistics, MOE, Peking University
[2] School of Software and Microelectronics, Peking University
[3] School of Computer Science, Peking University
[4] Huawei Technologies Ltd, China
 {wxylemon, shiinasama}@stu.pku.edu.cn    lisujian@pku.edu.cn

## Abstract

Reinforcement Learning from Verifiable Rewards (RLVR) has significantly improved the performance of large language models (LLMs) on tasks with gold ground truth, such as code generation and mathematical reasoning. However, its application to open-ended question answering (QA) remains challenging, primarily due to the absence of reliable evaluation and verifiable reward signals. This difficulty is further compounded by the limitations of existing evaluation paradigms. Previous approaches typically rely on human feedback or LLM-as-Judge strategies, which are costly, prone to reward hacking, and often fail to provide sufficiently discriminative or interpretable evaluation signals. To address these limitations, we introduce a schema for generating case-wise rubrics that are question-specific, content-based and stylistically sensitive, thereby evaluating both factual soundness and writing quality. Building on this schema, we propose QuRL (Open-Ended QA with Rubric-guided Reinforcement Learning), a framework that automatically mines rubrics for each question from easily accessible online sources and leverages them as reward signals. With these rubrics, QuRL employs the GRPO (Group Relative Policy Optimization) algorithm to guide the model in exploring the correct generation path. Extensive experiments on three different benchmarks show that our framework achieves significant improvements of total +17.0 points over a supervised fine-tuning baseline, demonstrating the effectiveness of rubric-guided reinforcement learning for open-ended QA.

## 1 Introduction

Over the past few years, Large Language Models (LLMs) have demonstrated impressive performance and achieved remarkable success in multiple NLP tasks. Building on these advancements, Reinforcement Learning from Verifiable Rewards (RLVR) has recently emerged as a powerful paradigm for further enhancing LLMs, exemplified by the success of DeepSeek-R1 and OpenAI's o-series. In RLVR, the model's reward comes from deterministic, rule-verifiable reward signals, which also enables RLVR to achieve significant improvements in tasks such as code and mathematics, as these domains possess consensus-based "gold answers". However, in practice, most real-world tasks do not provide clear, verifiable answers, leaving models without a straightforward source of reward feedback. A representative example is Open-Ended Question Answering (Open-Ended QA), where the absence of a single "gold answer" makes reliable evaluation particularly challenging.

Open-Ended QA is a task that is both challenging to answer and difficult to evaluate. Compared to closed-ended QA tasks such as mathematical or reasoning, Open-ended QA requires model responses not only to be factually accurate, but also to be fluently written and engaging enough to resonate with or capture readers' interests, thereby making human preferences the de facto gold standard for evaluation and learning. A widely adopted paradigm in open QA is Reinforcement Learning

---

[*]Equal contribution.
[†]Corresponding authors.

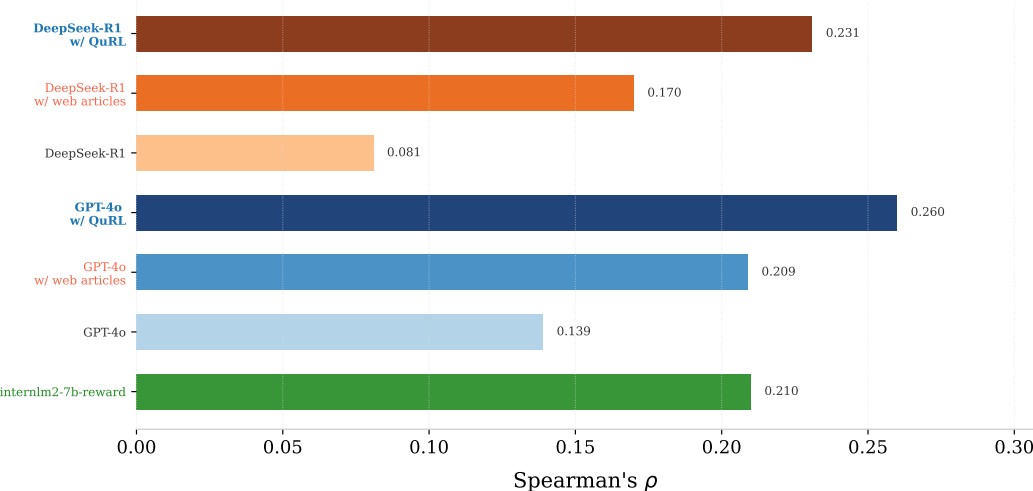

Figure 1: Incorporating web articles and QuRL-derived case-wise rubrics significantly enhances the alignment of LLM-as-a-Judge evaluations with human preferences.

from Human Feedback (RLHF) (Christiano et al., 2017), where annotators provide pairwise or scalar preference signals that are distilled into a reward model producing scalar scores. This reward model then serves as the supervision signal in reinforcement learning, guiding LLMs toward outputs that better align with human judgments. In our opinion, when annotators evaluate open-ended responses, they are essentially applying an implicit set of scoring rules (i.e., rubrics) to verify the quality of each answer. Here, we interpret the parameterized reward model trained in RLHF as a statistical approximation of evaluation rubrics, which encode the latent evaluation criteria with a scalar-valued function.

However, the parameterization of rubrics in RLHF often suffers from poor out-of-domain generalization and is vulnerable to reward hacking, since the underlying rubrics remain implicit and entangled within model parameters (Liu et al., 2024b; Wang et al., 2024). A natural remedy is to move from implicit to explicit supervision: if we can design case-specific rubrics as evaluation references for each question, the annotation task would no longer rely on hidden, parameterized standards. Instead, it would be grounded in clear, interpretable criteria, thereby mitigating the above issues and providing more stable guidance for reinforcement learning. In this way, evaluation shifts from depending on opaque reward models to verifiable rubrics, effectively extending RLVR beyond strictly verifiable domains to the open-ended QA setting. Then, a challenging question is how to obtain explicit guidance and create compact rubrics for learning. One straightforward approach is to employ human experts to author detailed rubrics, yet the prohibitive annotation costs involved make it impractical for large-scale training pipelines. Alternatively, we observe that the Internet already contains a wealth of human-authored materials related to open-ended questions including essays, articles, or forum discussions, which can serve as coarse-grained rubrics or inspirations for rubric construction. To validate the feasibility of the web sources, we randomly sampled 50 open-ended questions. For each question, we retrieved relevant web articles as evaluation prompt (w/ web articles) and guide the evaluation of the answers. We then generated 3 candidate responses for each question using GPT-4o (Hurst et al., 2024) and collected human evaluation scores based on the checklist described in Figure 7. Different LLM-as-a-Judge evaluation methods (i.e., DeepSeek-R1 and GPT-4o) w/o web articles are used to evaluate the responses and their correlation to human scores are compared. As shown in Figure 1, we can see that leveraging web articles as coarse-grained reference rubrics can improve the alignment between LLM evaluations and human annotator preferences, and explicit guidance leads to better performance than the reward-model-based approach (internlm2-7b-reward (Cai et al., 2024)) that captures human preferences. But directly incorporating raw web articles into evaluation introduces severe practical issues: the context length often exceeds 100k tokens, substantially inflating computational costs and limiting scalability. This further raises a new problem: **how we extract and distill from noisy web articles into information-dense and rubric-like signals that remain both effective and efficient for supervision**.

To address the above issue, we propose **QuRL** (Open-Ended Question Answering with Rubric-guided Reinforcement Learning), a framework that transforms human-authored web sources into case-wise rubrics and integrates them into RLVR as supplementary signals for answer verification. Specifically, QuRL first retrieves relevant human-authored text, distills them into concise meta-descriptions, and constructs rubric items with filtering to ensure discriminative quality. These case-wise rubrics are then used as structured reward signals in GRPO, enabling the model to internalize both content coverage and stylistic quality in a scalable and interpretable manner. By distilling noisy web articles into compact, discriminative rubrics, QuRL preserves interpretability while remaining computationally efficient. Importantly, these distilled rubrics yield stronger alignment with human judgments, as shown in Figure 1. Compared with existing paradigms, QuRL inherits the alignment benefits of RLHF while avoiding the opacity and instability of reward models, and at the same time preserves the simplicity and scalability of RLVR by eliminating the need for large-scale manual annotation or specialized reward model training.

Our contributions can be summarized as follows:

- We introduce QuRL, a framework that leverages internet text to construct case-wise rubrics as reward signals for open-ended question answering, thereby enabling RLVR in subjectively evaluated tasks. To the best of our knowledge, this is the first work to extend RLVR to the open-ended QA domain by utilizing fine-grained rubrics distilled from human-authored web sources.
- With the assistance of QuRL, we constructed a *QuRL-Train* dataset consisting of 800 Question–Rubric pairs, along with a *QuRL-Test* dataset of 400 entries that underwent human verification.
- Experimental results across three benchmarks (HelloBench, LongBench-Write, and QuRL-Test) demonstrate the effectiveness of our approach: when trained with GRPO under the QuRL framework, Qwen-2.5-7B achieves an average improvement of over +17.0 points compared to its supervised baseline.

## 2 RELATED WORK

**Open-Ended QA**. With the advancement of language model architectures (Chen et al., 2023; Zhu et al., 2023; Peng et al., 2024; Ding et al., 2024; An et al., 2024; Jin et al., 2024), their capabilities have gradually expanded from generating short responses to producing longer, open-ended answers. Open-Ended QA presents unique challenges compared to conventional closed-ended or extractive QA (Yang et al., 2018; Trivedi et al., 2022; Wang et al., 2023). One core difficulty lies in question ambiguity and the absence of a single correct answer. Previous attempts (Kantharaj et al., 2022) include extracting long descriptive passages from a full article to serve as a "gold answer" and then designing questions for evaluation in a manner similar to closed-ended QA. While this approach offers a referenceable standard answer, it also imposes relatively strict constraints on response evaluation (e.g., through ROUGE metrics). Given that the vast majority of open-ended QA tasks lack a gold answer, this method of dataset construction is inherently limited. With the improvement of large model capabilities, some studies (Que et al., 2024; Tan et al., 2024; Farzi & Dietz, 2024; Song et al., 2024; Hashemi et al., 2024; Xu et al., 2024; Biyani et al., 2024; Jain et al., 2023) have begun to deconstruct open-ended questions along multiple dimensions and employ LLM-as-a-Judge for evaluation. For example, HelloBench (Que et al., 2024) uses a fixed checklist for all questions, converting responses into scalar scores with some generalizability. However, our experiments show that this approach lacks discriminative power, suggesting that question-specific rubrics are needed to better distinguish response quality.

**Reinforcement Learning from Verifiable Rewards**. Reinforcement Learning with Verifiable Rewards (RLVR) has been adopted by DeepSeek-R1 (Guo et al., 2025) and OpenAI's o-series models (Jaech et al., 2024) due to its easily scalable training framework (Ma et al., 2025). However, in the field of open-ended QA where verifiable reference answers are unavailable, the common practice remains the use of RLHF to align with human preferences. Traditional RLHF relies on scalar reward models to obtain reward signals (Cai et al., 2024; Ouyang et al., 2022; Wu et al., 2023; Hu et al., 2025; Son et al., 2024), which suffer from weak interpretability and are vulnerable to reward hacking (Fu et al., 2025; Xu et al., 2025; Mahan et al., 2024; Chen et al., 2024). To mitigate these issues, some studies (Mahan et al., 2024; Liu et al., 2025; Gunjal et al., 2025; Huang et al., 2025) have introduced Generative Reward Model (GRM) as an extension of RLHF. Among them, Huang et al. (2025) is most related to our work. Their approach (has not been open-sourced

yet) constructs rubrics through a complex LLM-agent pipeline, without leveraging Internet resources. Besides, Previous studies (Li et al., 2025; Do et al., 2025) have also explored generating rubrics from questions. However, due to framework design limitations, their methods cannot be used for reward signal generation in RL processes and lack reliable evaluation in open-ended QA (with no publicly available reproducible code). Compared to methods mentioned above, our QuRL framework automatically mines case-wise rubrics from web sources, offering a lightweight, reproducible, and human-aligned reward signal for open-ended QA.

# 3   METHODOLOGY

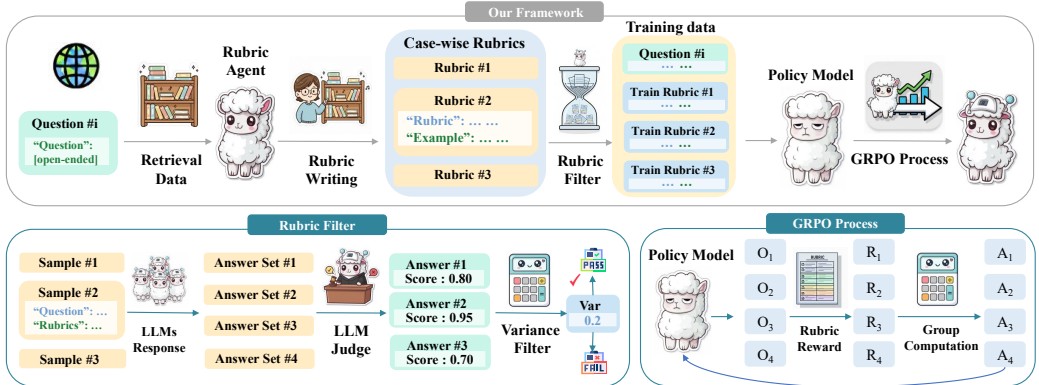

Figure 2: Overview of the QuRL framework. For each open-ended question, relevant human-authored materials are retrieved to guide rubric writing, producing case-wise rubrics with illustrative examples. A rubric filter ensures reliability by discarding inconsistent rubrics. The filtered rubrics are then used to score model responses, providing supervision signals for GRPO training.

## 3.1   RUBRICS GENERATION VIA INTERNET DATA

Results in Figure 1 show that incorporating relevant reference articles is beneficial when using LLMs as evaluators. However, directly leveraging raw, unfiltered web sources introduces substantial noise and computational overhead, limiting the practicality of this approach. We design the QuRL framework in Figure 2 to extract high-quality, case-wise rubrics from Internet sources and employ them as supervision signals in RL training.

- **Question-based Retrieval**   As shown in Figure 2, we first construct retrieval queries that may contain reference articles for answering the specific question. For the given open-ended question $q$, the search engine returns retrieved webpages $W = \{w_i \mid i = 1, 2, \ldots, N_w\}$ ranked by click-through rate based on the keywords, where $w_i$ denotes the text of the webpage with sequence number $i$, and $N_w$ is the maximum number of retrieved webpages. It aligns with our objectives since high click-through rates often indicate that the content is widely recognized and therefore of relatively high quality.

- **Meta-Description**   To handle the noisy and massive raw text from the retrieved webpages, we employ a lightweight and responsive model (Qwen2.5-7B) to generate concise *meta-descriptions*. Each meta-description is obtained by prompting the model to extract only the information that is relevant to answering the given open-ended question, while filtering out tangential details, advertisements noise (Appendix A.5). The guiding principle is to retain valuable and complete content segments, such as descriptions of key arguments, passages that provide informative context, transitional reasoning that bridges ideas, or well-crafted illustrative examples. Let $W = \{w_i \mid i = 1, 2, \ldots, N_w\}$ denote the set of retrieved webpages. We define a extraction function

$$f_D : W \to D, \quad D = \{d_i \mid i = 1, 2, \ldots, N_d\},$$

where $d_i = f_D(w_i)$ represents the distilled description corresponding to $w_i$, $N_d$ is the number of retained meta-descriptions. Each $d_i$ is thus a compact, information-dense representation that preserves semantically valuable content while discarding irrelevant noise.

> **Rubric Design Principles and Showcases**
>
> 1. **Content Focus**: Cover the essential argumentative and informational aspects extracted from meta-descriptions, ensuring answers are judged against the core content of the question.
>
> 2. **Writing Quality**: Evaluate human-like writing aspects such as clarity, logical flow, engagement, and stylistic richness, tied closely to the case.
>
> 3. **Case-Wise Specificity**: Avoid generic criteria; every scoring rule and illustrative example must be specific to the provided question.
>
> 4. **Meta-Description Referencing**: Leverage meta-descriptions as both argumentative references and stylistic exemplars for rubric construction.
>
> **Collected Rubrics:**
> Clarity and Logical Flow (2 points)
> - *Excellent (2 pts)*: Answer is structured with clear cause-and-effect reasoning, linking the Nobel Committee's criteria (e.g., foundational discovery vs. applied science) to the exclusion of cure developers. Example: "The Nobel Prize traditionally prioritizes foundational discoveries—like identifying HCV—over translational work. The 2020 award recognized the virus's discovery as the critical first step enabling later drug development, which often involves large, diffuse teams."
> - *Needs Improvement (0 pts)*: Disjointed or lacks transitions. Example: "Scientists found HCV. Drugs came later. Nobel didn't give prizes for drugs."
> ...

Figure 3: Illustration of the rubric design principles and example rubrics collected from meta-descriptions.

- **Rubrics Construction with Principles**  After obtaining meta-descriptions for each question, we find that they can provide guidance for response evaluation from two complementary perspectives. First, meta-descriptions often contain elaborations on a particular viewpoint derived from the question, which mirrors how human writers typically form several core stances from memory when composing answers to similar questions. This suggests that meta-descriptions can serve as argumentative references, providing concrete guidance on the key points that rubrics should emphasize when evaluating responses. Second, we observe that a common failure mode of current LLMs on open-ended questions lies in their misalignment with human writing styles—for example, producing enumerative but shallow discussions, lacking coherent transitions, or using language that is unengaging. In contrast, meta-descriptions, being distilled from human-authored texts, naturally preserve stylistic and rhetorical features that can serve as exemplars for high-quality writing. Based on these observations, we incorporate meta-descriptions into rubric generation along two dimensions: **content quality** (capturing core arguments and viewpoints) and **writing quality** (capturing human-like fluency, coherence, and expressiveness). We derived four rubric design principles from the above assumptions and, based on them, collected rubric examples as shown in Figure 3. Formally, we define a rubric construction function

$$f_{\mathrm{R}} : (q, D) \to R, \quad R = \{r_i \mid i = 1, 2, \ldots, N_r\},$$

where $R$ is the final set of case-wise rubrics $r_i$, $N_r$ is the total number of rubrics.

- **Rubrics Filter**  Since rubric generation is inherently stochastic, multiple sampling runs yield diverse candidate sets $\mathcal{R}(q) = R^{(1)}, \ldots, R^{(K)}$. We design a filtering mechanism (details in Appendix A.2) that (i) removes rubric sets lacking discriminative power across model responses, and (ii) consolidates rubrics that are consistently reproduced across samples. The resulting high-quality rubric set is denoted as $R^*(q)$, which will be used in subsequent process.

- **Collected Dataset with Rubrics**  Through the above construction process, we obtain a robust rubric set $R^*(q)$ for each question $q$. To build our dataset, we collected 1200 questions from the ten most popular topical domains. Among them, 400 questions together with their validated rubrics were manually double-checked, forming the test set *QuRL-Test* $= \{(q_i, R^*(q_i))\}_{i=1}^{400}$ (detailed in Appendix A.3), while the remaining 800 questions were used as the training set *QuRL-Train* $= \{(q_i, R^*(q_i))\}_{i=1}^{800}$, for subsequent reinforcement learning. Human annotators involved in this verification and labeling process were compensated at rates consistent with market standards, and each item was independently annotated by at least two annotators, with additional passes used to resolve substantial disagreements.

## 3.2 REINFORCEMENT LEARNING WITH CASE-WISE RUBRICS

With the construction of *QuRL-Train*, we proceed to the post-training stage to further align the model with human-preferred evaluation standards. The core idea is to utilize the case-wise rubrics $R^*(q)$ as structured reward references, guiding the optimization of the policy model through Group Relative Policy Optimization (GRPO).

**Rubric-based Reward Modeling.** For each training tuple $(q_i, R^*(q_i)) \in$ *QuRL-Train*, given a candidate answer $o$ sampled from the policy $\pi_\theta$, we query a judge model $\text{LLM}_{\text{reward}}$ to produce a detailed evaluation text $y$, which contains rubric-wise judgments:

$$y = \text{LLM}_{\text{reward}}(q_i, o, R^*(q_i)). \tag{1}$$

A deterministic parser $f$ then extracts numerical rubric-level scores from $y$ and normalizes their sum into the range $[0, 1]$, yielding the final reward

$$R(o \mid q_i, R^*(q_i)) = f(y). \tag{2}$$

**Group Relative Policy Optimization.** We adopt GRPO (Guo et al., 2025) as the reinforcement learning algorithm. For each $(q_i, R^*(q_i)) \in$ *QuRL-Train*, we sample $N$ candidate answers $\{o_1, o_2, \ldots, o_N\}$ from the policy $\pi_\theta$, and compute the corresponding rewards

$$R_j = R(o_j \mid q_i, R^*(q_i)).$$

We then normalize these rewards to obtain the relative advantage of each sampled answer:

$$A_j = \frac{R_j - \text{mean}\{R_1, R_2, \ldots, R_N\}}{\text{std}\{R_1, R_2, \ldots, R_N\}}. \tag{3}$$

Finally, the policy is updated under the GRPO objective:

$$\begin{aligned}
J_{\text{GRPO}}(\theta) = \mathbb{E}_{(q_i, R^*(q_i)) \sim \textit{QuRL-Train}, \{o_j\}_{j=1}^N \sim \pi_{\theta_{\text{old}}}} \\
\left[ \frac{1}{N} \sum_{j=1}^N \min\left( \frac{\pi_\theta(o_j \mid q)}{\pi_{\theta_{\text{old}}}(o_j \mid q)} A_j, \right. \right. \\
\left. \text{clip}\left( \frac{\pi_\theta(o_j \mid q)}{\pi_{\theta_{\text{old}}}(o_j \mid q)}, 1 - \varepsilon, 1 + \varepsilon \right) A_j \right) \\
\left. - \beta D_{\text{KL}}\left( \pi_\theta \,\|\, \pi_{\text{ref}} \right) \right],
\end{aligned} \tag{4}$$

where $\varepsilon$ is the clipping hyper-parameter and $\beta$ controls the KL divergence penalty with respect to a reference policy $\pi_{\text{ref}}$. Through GRPO, the case-wise rubrics directly influence the reward signal and thereby control the policy parameter updates. Finally, the post-trained model learns to internalize the rubric-based evaluation criteria and align its outputs with the qualities of human-authored writing.

## 4 EXPERIMENTS

### 4.1 EXPERIMENTAL SETUPS

**Benchmark Setup.** We evaluate model performance on three evaluation settings:

**(1) HelloBench (Que et al., 2024)** HelloBench is a large-scale, open-ended benchmark covering diverse topical domains. Its open-ended QA task adopts a five-dimension checklist aligned with human judgments, and derives a composite score via regression-fitted weights.

**(2) LongBench-Write (Bai et al., 2024)** LongBench-Write contains 120 varied prompts for long-form writing evaluation. Following its original setup, we use the paper's quality score to assess model outputs. This paper also introduces a post-training model **LongWriter-9B-DPO** that has been further enhanced to improve its text generation capabilities, which will serve as a comparison baseline in the subsequent experiments.

| Models | Avg | | HelloBench | QuRL-Test | LB-Write |
|---|---|---|---|---|---|
| | Score | Len | | | |
| *Proprietary LLMs* | | | | | |
| GPT-4o (GPT-4o, 2024) | 64.7 | 1096 | 46.0 | **80.8** | 67.2 |
| Gemini-2.5-Pro (Comanici et al., 2025) | **70.4** | 1137 | **69.2** | 65.9 | **76.1** |
| Gemini-2.5-Flash (Comanici et al., 2025) | 62.3 | 1113 | 48.4 | 64.8 | 73.6 |
| Doubao-Seed-1.6 | 40.2 | 996 | 24.1 | 31.2 | 65.2 |
| *Open-source LLMs* | | | | | |
| DeepSeek-R1 (Guo et al., 2025) | **62.4** | 735 | 32.8 | **80.4** | 74.0 |
| DeepSeek-V3 (Liu et al., 2024a) | 59.1 | 742 | 28.1 | 70.8 | **78.4** |
| Qwen2.5-7B-Instruct (Yang et al., 2024) | 28.3 | 923 | 20.8 | 26.2 | 37.8 |
| Qwen2.5-72B-Instruct (Yang et al., 2024) | 42.3 | 853 | **34.4** | 41.2 | 51.2 |
| Llama-3.1-8B-Instruct (Dubey et al., 2024) | 23.7 | 997 | 25.6 | 33.2 | 12.4 |
| Llama-3.1-70B-Instruct (Dubey et al., 2024) | 31.9 | 1028 | 28.0 | 46.4 | 21.2 |
| *Post-enhanced LLMs* | | | | | |
| LongWriter-9B-DPO Bai et al. (2024) | 27.5 | 2164 | 24.8 | 16.0 | 41.6 |
| Qwen2.5-7B-Coldstart | 34.1 | 807 | 26.4 | 35.6 | 40.4 |
| Qwen2.5-7B-SFT | 42.3 | 1214 | 38.0 | 41.6 | 47.2 |
| Qwen2.5-7B-QuRL | **59.3** | 916 | **56.4** | **62.4** | **59.2** |
|   *-w/ rlhf reward model* | 47.7 | 879 | 44.8 | 50.4 | 48.0 |
|   *-w/o rubrics filter* | 52.2 | 951 | 48.1 | 54.4 | 54.0 |
|   *-w/o rubrics* | 44.0 | 841 | 40.8 | 45.9 | 45.2 |
|   *-w/o web information* | 48.9 | 847 | 45.2 | 53.6 | 47.9 |

Table 1: Main Results of LLMs across multiple benchmarks. The "Avg" column represents the average score ("Score") and the average response length ("Len") for each model. To ensure fairness, the scores from different benchmarks are normalized like the score from HelloBench (Que et al., 2024). "LB-Write" is short for LongBench-Write. The ablation results listed under 'Post-enhanced LLMs' represent single ablations. We use 'w/' as an abbreviation for 'with' and 'w/o' for 'without' in these variants.

**(3) QuRL-Test (ours)** Unlike HelloBench's fixed dimensions, *QuRL-Test* introduces fine-grained, case-wise rubrics tailored to each question, capturing distinct writing styles and content emphases.

**Comparison of LLMs.** To comprehensively evaluate performance across different model families, we include both proprietary and open-source large language models in our study. For *proprietary* models, we evaluate **GPT-4o**, **Doubao-Seed-1.6**, **Gemini-2.5-Pro**, and **Gemini-2.5-Flash**. For *open-source* models, we include **LLaMA-3.1-8B,70B-Instruct** (Dubey et al., 2024), **Qwen2.5-7B,72B-Instruct** (Yang et al., 2024), **DeepSeek-R1** Guo et al. (2025), and **DeepSeek-V3** Liu et al. (2024a). These models cover a broad spectrum of parameter scales, training paradigms, and accessibility levels, enabling us to compare QuRL-enhanced training with both state-of-the-art proprietary systems and widely used open-source baselines.

**Implementation Details.** We adopt *QuRL-Train* as our training corpus. Following common practice in reinforcement learning with LLMs, we first perform a cold-start supervised fine-tuning stage to facilitate model adherence to the <think>–</think> and <answer>–</answer> format. Specifically, we distill 64 instruction–response pairs from **DeepSeek-R1** and conduct cold-start with a learning rate of $1\mathrm{e}{-6}$, batch size of 16, and 2 training epochs. After initialization, we adopt the GRPO algorithm for alignment. During RL training, we set the learning rate to $1\mathrm{e}{-6}$ and run for 2 epochs. For each question, the policy samples 8 candidate responses, and rewards are computed according to the rubric-based evaluation described above. The global batch size is fixed at 32. For fair comparison, we report the best performance achieved across the two epochs. All training procedures are conducted on 8 A100 GPUs.

## 4.2 MAIN RESULTS

Table 1 presents the performance of LLMs across the evaluated datasets.

**RLVR as the Most Effective Alignment Strategy** We compare against several baselines: *Qwen2.5-7B-SFT*, trained on DeepSeek-R1 (Guo et al., 2025) responses from QuRL-Train; and *Qwen2.5-7B-QuRL w/ rlhf reward model*, which replaces rubric-based supervision with scalar rewards from a trained RLHF model (internlm2-7b-reward (Cai et al., 2024)) for GRPO training. As shown in Table 1, SFT achieves only 42.3 on average, falling short of RL-based methods. RLHF further improves performance to 47.7, but remains limited by fragile scalar rewards. In contrast, our rubric-based RLVR reaches 59.3, a clear margin of +17.0 over SFT and +11.6 over RLHF on average across the three benchmarks (HelloBench, LongBench-Write, and QuRL-Test). These results confirm that RLVR provides more stable and discriminative reward signals, yielding consistently stronger performance across all benchmarks.

**Rubrics and Human Writing Materials as Key Drivers of RLVR Success** Ablation results highlight the crucial role of both rubrics and human-authored writing materials in RLVR's success on open-ended QA. Specifically, the *w/o rubrics* variant removes rubric-based supervision and instead relies on a five-dimension scoring prompt for LLM-as-a-Judge evaluation as shown in Figure 7, while the *w/o web information* variant excludes distilled human-authored texts, meaning the model generates case-wise rubrics without referencing any external materials. In addition, the *w/o rubrics filter* variant retains rubric-based supervision but discards the filtering mechanism that removes noisy rubric items, leading to less reliable supervision. Removing rubrics reduces the average score from 59.3 to 44.0, removing the filter results in 52.2, while removing human-authored materials lowers it to 48.9. These findings indicate that case-wise rubrics provide precise, verifiable evaluation signals, the filtering step further enhances their reliability, and human-authored meta-descriptions enrich content coverage. Together, they enable RLVR to capture both argumentative quality and writing style.

**Performance Trends in Different LLMs** We observe that among proprietary models, *Gemini-2.5-Pro* demonstrates consistently superior performance across all three benchmarks, achieving the highest average score of **70.4**. Meanwhile, in the open-source category, model size emerges as a critical factor: larger variants such as *Qwen2.5-72B* and *LLaMA-3.1-70B* clearly outperform their smaller counterparts. Notably, the *DeepSeek* series, with its near-700B parameter scale, establishes an absolute advantage among open-source models, reaching 62.4 and 59.1 on average and surpassing other open-source baselines by a considerable margin. These observations highlight both the strong competitiveness of proprietary SOTA models and the decisive role of parameter scaling in shaping open-source model performance. Within the post-enhanced category, the *LongWriter-9B-DPO* shows competitive performance only on the LongBench-Write benchmark. We attribute this to its specialized training objective of producing extended outputs: its responses average over 2000 words, substantially longer than other models. While such length optimization enables the generation of lengthy narratives, it does not guarantee alignment with human preferences regarding answer quality. As a result, *LongWriter-9B-DPO* fails to generalize beyond LongBench-Write, performing poorly on HelloBench and QuRL-Test. It is worth noting that *Qwen2.5-7B-QuRL*, supervised with only 800 case-wise rubrics, achieves performance on par with *DeepSeek-V3*, indicating the efficiency of our training paradigm. Moreover, by examining the average output length, we find that QuRL lies in the middle range among all models, suggesting that its strong performance does not result from artificially inflating response length to game the evaluation metrics.

### 4.3 HUMAN EVALUATION CONSISTENCY

|  | QuRL | HelloBench | LongWriter | InternLM2 | LLM-as-a-Judge |
|---|---|---|---|---|---|
| **Spearman's $\rho$** | 0.31 | 0.20 | 0.11 | 0.22 | 0.08 |
| $p$-**Value** | 8.29e-6 | 3.36e-3 | 2.44e-2 | 1.89e-4 | 5.31e-2 |

Table 2: Consistency between automatic evaluation methods (all use GPT-4o as judge model) and human judgments on 200 GPT-4o responses to HelloBench questions. We report Spearman's $\rho$ correlation coefficients and the corresponding significance levels ($p$-values).

Evaluating open-ended responses is inherently challenging since no single gold-standard answer exists for reference. To assess the reliability of our evaluation based on the QuRL rubric, we follow previous work (Que et al., 2024) and conduct a human evaluation consistency analysis. Specifically, we generated responses to 200 HelloBench questions using GPT-4o and then scored them under five evaluation settings: (1) **QuRL**, using case-wise rubrics tailored to each question; (2) **HelloBench**, which aggregates five dimension scores with learned weights; (3) **LongWriter**, which uses its quality

score definition; (4) **InternLM2**, using the internlm2-7b-reward (Cai et al., 2024) as the reward model; and (5) **LLM-as-a-Judge**, which uses the prompt as shown in Figure 7. For human reference, annotators were asked to rate the same set of responses using the identical scheme as in the LLM-as-a-Judge setting. Finally, we report the Spearman's rank correlation between each automatic evaluation method and human ratings as shown in Table 2. A higher $\rho$ indicates stronger agreement, while a lower $p$-value indicates greater significance. The results indicate that QuRL achieves the strongest correlation with human judgments ($\rho = 0.31$, $p < 10^{-5}$), substantially outperforming other methods. This advantage demonstrates the reliability of case-wise rubrics and also explains why models trained with QuRL-Train exhibit significant performance gains across benchmarks.

## 4.4 RUBRIC SCORING SCHEMES

| Scoring Scheme | Avg | HelloBench | QuRL-Test | LB-Write |
|---|---|---|---|---|
| Fixed-maximum | **59.3** | 56.4 | 62.4 | 59.2 |
| Free-form | 50.4 | 47.2 | 53.2 | 50.8 |
| Judge-based | 52.4 | 49.2 | 56.8 | 51.2 |

Table 3: Comparison of different rubric scoring schemes across benchmarks.

While the previous section highlights the effectiveness of case-wise rubrics, QuRL ultimately relies on these rubrics to generate reward signals during training. A key design question is therefore: *how should rubrics be translated into final scores*? Different scoring schemes may lead to different supervision strengths and inductive biases, and exploring these alternatives sheds light on how rubrics can best guide model learning. We consider three schemes as following:

**(1) Fixed-maximum scoring.** Given a pre-defined maximum score (e.g., $max\_points = 10$), the model is required to autonomously assign partial scores to each rubric item, ensuring that the total sums up to the fixed maximum.

**(2) Free-form scoring with normalization.** The model is allowed to freely decide the total score and the score assignment across rubric items. The total score is normalized to the interval $[0, 10]$.

**(3) Judge-based binary scoring.** Instead of assigning scores, the model only generates the rubrics. A separate judge model then determines whether each rubric is satisfied (binary decision). The final score is computed as the proportion of satisfied rubrics, normalized to $[0, 10]$.

As shown in Table 3, the Fixed-maximum scheme achieves the strongest performance across all benchmarks, and thus we adopt it as the default setting in our main results. However, different types of questions may call for different scoring allocations. For instance, in a question like "What will human life in space look like in the future?", concrete examples and vivid descriptions serve as key strengths, whereas for a question such as "How should we interpret Socrates' philosophy?", originality and creative insights become the most decisive factors. Designing more flexible scoring schemes that better approximate human annotators therefore remains an important direction for future.

## 4.5 CASE STUDY

To illustrate how rubrics function as supervision signals, Figure 4 presents a case study comparing model responses before and after QuRL training (showing partial excerpts for brevity). The initial answer, while coherent, remained abstract and generalized, offering limited personal depth. After training with rubric-guided reinforcement learning, the response became more vivid and persuasive: it incorporated concrete references (e.g., 1984), demonstrated perspective shifts on justice and freedom, and provided reflective insights on emotional resilience. This improvement directly aligns with the case-wise rubrics: criteria such as "Specific Examples" and "Emotional/Perspective Shifts" encouraged the model to enrich content with concrete illustrations and self-reflection, while "Clarity and Logical Flow" guided smoother organization of ideas. We observe that through the amplifying and suppressing effects of rubric-based rewards, the model was steered toward producing higher-quality answers. In Section A.4, we provide additional case studies for a more detailed analysis.

| Before Train vs After Train | Case-Wise Rubrics |
|---|---|
| **Question:** What changes has years of reading brought to you?

**Answer Before Train:** Years of reading have profoundly transformed my intellectual, emotional, and psychological landscape in ways that are both subtle and far-reaching…This immersion in diverse narratives has fostered a greater sensitivity to the struggles, joys, and motivations of others, making me more attuned to the emotional undercurrents in my relationships…

**Answer After Train:** Reading over the years has completely reshaped how I see the world and connect with others...Books like 1984 didn't just entertain me—they challenged my views on justice and equality and made me reflect on the dangers of absolute control and the fragility of personal freedom. That story sharpened my awareness of the value of truth and independent thought, lessons that continue to influence how I interpret the world around me…On a deeper level, reading also nurtured my emotional resilience. When life got tough, picking up a book became my way of finding comfort and processing complex feelings. Whether it was a novel or a piece of non-fiction, books became a therapeutic tool, guiding me through stressful times… | **Part 1: Writing Quality**
**1.1 Clarity and Logical Flow (2 points)**
Excellent (2 pts): The response is structured with smooth transitions between ideas, such as:"Initially, reading expanded my vocabulary, but over time, it also reshaped how I empathize with others—like how a novel's protagonist taught me to see grief through a stranger's eyes."
Needs Improvement (0–1 pt): Ideas are disjointed or lack progression, e.g., "Reading gave me new words. I also like fiction. Books are fun."
…
**Part 2: Content Coverage**
…
**2.2 Emotional/Perspective Shifts (1.5 points)**
Excellent (1.5 pts): Reflects on empathy or worldview changes with examples. "Memoirs like Educated made me question my biases about education and privilege."
Sufficient (1 pt): Briefly notes emotional impact.
Missing (0 pts): Omits emotional or perspective changes.
…
**2.5 Specific Examples (1 point)**
Excellent (1 pt): Cites titles, genres, or memorable passages. "Sapiens reshaped my understanding of human history, while The Midnight Library influenced my approach to regret."
Missing (0 pts): Lacks concrete references.
…
**2.6 Personal Reflection (1 point)**
Excellent (1 pt): Connects reading to self-identity or long-term growth. "Reading revealed my love for storytelling, leading me to start a blog."
Missing (0 pts): Superficial or impersonal response. |

Figure 4: Case study showing how rubric-guided training improves answer quality by encouraging specific examples, reflective depth, and clearer structure.

## 5  CONCLUSION AND FUTURE WORK

In this work, we introduced QuRL, a case-wise rubric-driven framework for aligning LLMs with human preferences on open-ended questions. By distilling case-wise rubrics from human-authored materials and integrating them into the RLVR paradigm, QuRL provides fine-grained and verifiable supervision signals that significantly improve evaluation reliability and model alignment. Experiments across multiple benchmarks demonstrate that QuRL achieves competitive performance with state-of-the-art systems, while human consistency analysis confirms the robustness of rubric-based evaluation. Our findings highlight the effectiveness of rubric-guided reinforcement learning and open promising directions for future research on controllable and preference-aligned LLM training.

As an extension of RLVR, QuRL opens several promising directions. Beyond aligning responses with human preferences, future work may explore finer-grained control over output length, where simple prompting is insufficient and stronger supervision is needed without sacrificing quality (Que et al., 2024; Bai et al., 2024). Another important avenue is assessing the safety of web-derived rubrics, since malicious or biased content could be injected and provide harmful supervision.

## ETHICS STATEMENT

This work relies on publicly available web documents and crowd-sourced annotations for evaluation and rubric verification. All data collection and usage followed the terms of service of the corresponding providers, and no personally identifiable or sensitive information was intentionally collected. Details of the annotation procedures and quality control are described in the dataset construction section and Appendix A.3.

## REPRODUCIBILITY STATEMENT

To ensure reproducibility, we have cleaned and released a part of core code used for rubrics collected at the following link: `https://anonymous.4open.science/r/QuRL-4EDF`.

ACKNOWLEDGEMENT

We thank the anonymous reviewers for their helpful comments on this paper. This work was partially supported by National Natural Science Foundation of China projects (No. 62476010 and 92470205).

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

# A APPENDIX

## A.1 THE USE OF LARGE LANGUAGE MODELS

Beyond evaluating the capabilities of different LLMs in our experiments, we only employed LLMs for post-processing, specifically to check for typos and grammatical errors in the written text.

## A.2 DETAILS FOR RUBRICS FILTER

Rubric construction process based on $f_R$ inherently depends on the sampling behavior of the generation model. Since the model may produce different phrasings and emphasize different aspects across sampling runs, applying $f_R$ multiple times to the same question $q$ with the same set of meta-descriptions $D$ can yield distinct rubric sets. Formally, we write

$$R^{(k)} = f_R^{(k)}(q, D), \quad k = 1, 2, \ldots, K,$$

where $R^{(k)} = \{r_i^{(k)} \mid i = 1, 2, \ldots, N_r^{(k)}\}$ denotes the $k$-th sampled rubric set, and the total number $K$ is set to 10 in practice. Thus, for a fixed question $q$, we obtain a family of rubric sets

$$\mathcal{R}(q) = \{R^{(1)}, R^{(2)}, \ldots, R^{(K)}\}.$$

This naturally raises the next question: given the family of rubrics sets $\mathcal{R}(q)$ generated for a single question $q$, how can we determine the final, reliable rubric set to be used for evaluation? To address this, we introduce a filtering mechanism that consolidates diverse rubric candidates into a consistent and high-quality set. The objective of our filtering mechanism is twofold: (1) to ensure that rubrics possess sufficient discriminative power across responses of different quality, and (2) to retain only those rubrics that show consensus, i.e., rubrics that consistently appear with identical or semantically similar formulations across multiple sampling runs.

For the first objective, we construct a response set for each question $q$ by sampling answers from multiple strong LLMs. Specifically, we use four different models (GPT-4o, Gemini-2.5-Pro, Qwen2.5-72B, DeepSeek-R1), and from each model we extract 5 responses. This yields a total of

$$\mathcal{A}(q) = \{a_1, a_2, \ldots, a_{20}\}$$

distinct responses for question $q$. We then employ GPT-4o as the judge model to evaluate these responses under each rubrics set $R^{(k)} \in R(q)$. Formally, for rubrics set $R^{(k)}$, we obtain a score vector

$$\mathbf{s}(R^{(k)}) = \big(s_1(R^{(k)}), s_2(R^{(k)}), \ldots, s_{20}(R^{(k)})\big),$$

where $s_j(R^{(k)})$ denotes the evaluation score assigned to response $a_j$ according to rubrics set $R^{(k)}$. After obtaining the score distribution for each rubrics set, we remove rubrics that lack discriminative power. Concretely, if the score variance is too small relative to the full scoring range, the rubrics set fails to differentiate between high- and low-quality responses. We compute the variance

$$\sigma(R^{(k)}) = \frac{1}{20} \sum_{j=1}^{20} \big(s_j(R^{(k)}) - \bar{s}(R^{(k)})\big)^2,$$

where $\bar{s}(R^{(k)})$ is the mean score of $R^{(k)}$. Let $\Delta$ denote the total score range (i.e., the maximum possible score minus the minimum possible score). We filter out $R^{(k)}$ if

$$\sigma(R^{(k)}) < \alpha \cdot \Delta,$$

where $\alpha \in (0, 1)$ is a threshold hyperparameter. Based on the empirical analysis of human annotators' scoring distributions, we calibrated the threshold $\alpha$ such that rubrics with insufficient discriminative power are reliably excluded while preserving those aligned with human judgment. Concretely, we set $\alpha = 0.15$ which corresponds to the lower bound of variance observed among rubrics that human annotators consistently regarded as discriminative.

For the second objective, we then employ a summarization model (in practice, GPT-4o) to consolidate the rubric sets that pass the discriminative power test, prioritizing rubrics with higher occurrence frequency across samples and merging semantically similar ones with the instruction shown in Figure 6. The final rubric set is obtained as $R^*(q)$.

| Category | Count |
|----------|-------|
| Science | 40 |
| Technology | 40 |
| Write | 40 |
| Food | 40 |
| Movie | 40 |
| Book | 40 |
| Sport | 40 |
| Health | 40 |
| Travel | 40 |
| Music | 40 |
| **Total** | **400** |

Table 4: Category distribution of our dataset. Each of the ten categories contains 40 samples, totaling 400 instances.

## A.3 DATASET SUPPLEMENTARY MATERIALS

We followed the ten most frequent topical categories in HelloBench (Que et al., 2024) to guide our question collection, drawing sources from both Zhihu and Reddit. Unlike HelloBench, we enforced a strictly balanced distribution of 40 questions per category and carefully curated question quality. Specifically, we manually reviewed all questions to avoid over-concentration on narrow subtopics and to ensure diversity within each category. For instance, in HelloBench, the "Sport" category contains multiple consecutive questions focused only on golf, which may limit coverage. To further guarantee openness, we filtered out questions that could be trivially answered in one or two sentences. In total, we obtained 400 test questions. For each question, we collected rubrics using our proposed method and conducted human verification to ensure correctness, appropriate scoring, and fine-grained evaluation dimensions. Erroneous rubrics were regenerated until satisfactory. The resulting benchmark is referred to as the **QuRL-Test** dataset.

## A.4 CASE STUDY

We include several case studies that demonstrate the significant enhancements brought by QuRL training. Improvements—such as greater specificity, clearer organization, and deeper insight are highlighted in the text for clarity. Red highlights indicate shortcomings before training, while green highlights mark the improved or enriched parts after training.

---

**Case 1**

**Question:** What items do people typically bring with them when traveling by plane? How can one securely store personal belongings such as a phone or wallet during a flight?

**Answer Before Train:** When traveling by plane, it's essential to pack the right items to ensure a comfortable and stress-free journey. Here's a list of common items that people typically bring with them:
1. **Travel Documents**: Passport, ID, boarding pass, and any other necessary travel documents.
2. **Clothing**: Appropriate clothing for the destination, including layers for different weather conditions.
3. **Toiletries**: Travel-sized toiletries, such as toothbrush, toothpaste, shampoo, and conditioner.
4. **Electronics**: Phone, charger, headphones, and any other electronic devices.
5. **Snacks and Water**: Healthy snacks and plenty of water to stay hydrated during the flight.
6. **Reading Material**: Books, magazines, or e-readers to pass the time.
7. **Comfort Items**: Blankets, pillows, or eye masks to make the flight more comfortable.

---

8. **Medications**: Any necessary medications and a first-aid kit.
9. **Travel Insurance**: Proof of travel insurance in case of any unforeseen circumstances.
10. **Cash and Credit Cards**: Some cash and a credit card for emergencies or in case of lost luggage.

To securely store personal belongings such as a phone or wallet during a flight, consider the following tips:

1. **Use a Secure Bag**: Choose a bag with multiple compartments and zippers to keep your belongings organized and secure.
2. **Keep Valuables Close**: Keep your phone, wallet, and other valuables in a pocket or a small bag that you can easily access and keep close to you.
3. **Use a Travel Wallet**: A travel wallet with multiple compartments can help keep your cash, cards, and other documents organized and secure.
4. **Zip Up Your Bag**: Always zip up your bag when not in use to prevent anyone from accessing your belongings.
5. **Avoid Leaving Your Bag Unattended**: Never leave your bag unattended, even for a moment, as it can be easily stolen or tampered with.
6. **Consider a Travel Lock**: A travel lock can help secure your bag and prevent it from being opened without your permission.
7. **Keep Important Documents Safe**: Keep important documents, such as your passport and boarding pass, in a secure pocket or bag that you can easily access.

By following these tips, you can ensure that your personal belongings are safe and secure during your flight.

**Answer After Train:** When traveling by plane, passengers typically pack a variety of items to ensure a comfortable and organized journey. Essential documents such as identification (IDs), boarding passes, and travel insurance policies are usually the top priorities, as they are required for security checks and boarding. Many travelers also bring personal electronics like smartphones, tablets, and laptops, along with chargers and adapters, especially for international flights where voltage and plug types vary. Clothing is a significant portion of the luggage, including layers for different climates, comfortable shoes, and attire suitable for the destination's culture or weather. Snacks and water are common, as in-flight meals may not meet personal dietary preferences, and the aircraft's environment can be dehydrating. Some travelers also carry medications, travel-sized toiletries, and entertainment options like books or movies to pass the time.

Securing personal belongings during a flight is a critical aspect of travel safety. Airlines generally recommend that passengers keep essential valuables—such as phones, wallets, passports, and travel documents—on their person or in a small bag placed under the seat in front of them for easy access and security. Larger carry-on items can be stored in the overhead bin, but it is not advisable to leave valuables there, since access is less controlled and items may be misplaced. The seat pocket can be used for temporary storage of small items during the flight, but travelers should check it carefully before leaving the aircraft to avoid forgetting anything. It's crucial to avoid leaving belongings unattended at any point, as airports and planes are high-risk areas for theft. For example, during layovers or when using the restroom, travelers should keep their bags within sight or consider using the aircraft's storage options. Additionally, digital items like passports and tickets should be backed up on cloud services or printed copies, in case of technical issues or loss.

To enhance the travel experience, some tips include packing a small first-aid kit, earplugs, and an eye mask for a better sleep, especially on long flights. Staying hydrated and avoiding alcohol during takeoff and landing can also reduce ear discomfort. For international flights, it's wise to research cultural norms and dress codes at the destination to avoid cultural misunderstandings. Travelers with special needs, such as dietary restrictions or mobility issues, should notify the airline in advance and bring necessary medications or equipment. By following these guidelines, passengers can enjoy a smoother, more secure, and stress-free flight, focusing on the journey rather than the logistics.

**Rubrics:**
Part 1: Writing Quality (3 points total)

1.1 Clarity and Logical Flow (2 points)
Excellent (2 pts): Answer transitions smoothly between packing essentials and security tips. Example: "While travelers typically pack electronics and documents in carry-ons, securing these items requires strategic storage—like using a crossbody bag for quick access or a hidden pouch to deter theft."
Needs Improvement (0 pts): Disjointed statements lack cause/effect. Example: "People bring phones. Wallets are important. Use a lock."

1.2 Engagement and Stylistic Flair (1 point)
Excellent (1 pt): Vivid language or analogies enhance readability. Example: "A money belt is the unsung hero of air travel—nestled under clothing, it guards cash and passports like a silent sentinel."
Needs Improvement (0 pts): Dry, textbook-like recitation. Example: "Phones and wallets are commonly carried. Store them securely."

Part 2: Content Coverage (7 points total)

2.1 Typical Travel Items (1 point)
Excellent (1 pt): Lists 5+ common items (e.g., phone, wallet, passport, headphones, toiletries) and explains their importance.
Sufficient (0.5 pt): Mentions 3–4 items without context.
Missing (0 pts): Omits key items.

2.2 Secure Storage Methods (2 points)
Excellent (2 pts): Details 3+ methods (e.g., money belts, locked pouches, seat-pocket avoidance) with pros/cons.
Sufficient (1 pt): Lists 1–2 methods superficially.
Missing (0 pts): No storage solutions discussed.

2.3 Compliance with Rules (1 point)
Excellent (1 pt): Mentions following airline or airport security rules in general.
Sufficient (0.5 pt): Alludes to compliance vaguely.
Needs Improvement (0 pts): Ignores compliance.

2.4 Theft Prevention (1 point)
Excellent (1 pt): Gives clear anti-theft tips (e.g., avoid leaving items unattended, avoid storing valuables in overhead bins, keep bag within sight).
Sufficient (0.5 pt): Generic tip like "keep valuables close."
Missing (0 pts): No anti-theft measures.

2.5 Accessibility During Flight (1 point)
Excellent (1 pt): Explains how to balance security with in-flight access (e.g., under-seat personal items vs. locked compartments).
Sufficient (0.5 pt): Mentions accessibility without strategy.
Missing (0 pts): Omits accessibility.

2.6 Emergency Preparedness (1 point)
Excellent (1 pt): Suggests backups (e.g., photocopies of IDs, split cash storage) for lost items.
Sufficient (0.5 pt): Mentions emergencies vaguely.
Missing (0 pts): No contingency plans.

Case 2

**Question:** At what point does a person get "old"? Is it simply health-related factors, certain attitudes, beliefs, and regrets, or can it happen at any age?

**Answer Before Train:**The question of when a person becomes "old" is inherently subjective, as it intertwines biological, psychological, social, and cultural dimensions, making it impossible to pinpoint a single, universal answer. Age is not merely a chronological marker but a dynamic construct shaped by health, attitude, societal expectations, and personal identity.Biologically, aging is a process influenced by genetics, lifestyle, and environmental factors, with some individuals retaining vitality well into their 70s or 80s, while others may experience physical decline earlier. However,the transition into "old age" is often marked by the onset of age-related health issues such as arthritis, vision or hearing loss, cognitive decline, or chronic conditions like diabetes or heart disease. These milestones can trigger a sense of vulnerability or dependence, contributing to a psychological shift toward viewing oneself as no longer youthful or vigorous.

Psychologically, the perception of aging is deeply tied to self-perception and life stage. Many people associate "old age" with retirement, a milestone that liberates time but may also signal the end of a professional identity or routine. This transition can evoke feelings of loss or redundancy, particularly if one has not cultivated new interests or social roles.Additionally, societal norms and media portrayals of aging—often emphasizing decline or frailty—can influence how individuals internalize their age. For instance, in cultures that value youthfulness, older adults may feel pressure to maintain a youthful appearance or lifestyle, whereas in others, respect for elders is a cultural norm, delaying the psychological acceptance of aging. Personal regrets, such as unfulfilled career aspirations or strained relationships, can also contribute to a sense of existential aging, where one feels they have missed opportunities or failed to live up to their potential.

Ultimately, the point at which someone feels "old" is a complex interplay of internal and external factors. It is not a fixed age but a fluid state that evolves over time. While health crises or significant life events like the death of a spouse or loss of independence can accelerate this process, others may age gracefully, maintaining mental clarity, social engagement, and a positive outlook. The key takeaway is that aging is a multifaceted journey influenced by individual resilience, adaptability, and the cultural narratives we internalize. Recognizing aging as a spectrum rather than a binary state allows for a more compassionate and nuanced understanding of this natural process.

**Answer After Train:**The question of when a person "gets old" is inherently complex, as the term "old" is both a biological and a social construct, deeply intertwined with individual perception, cultural context, and life experiences. Age, in its most basic form, is a chronological measurement, but the experience of aging is far more nuanced. While biological age relates to the physical and physiological changes that occur over time, such as the shortening of telomeres, decline in organ function, and accumulation of cellular damage, these markers are not uniform across individuals. For example, a 60-year-old with a healthy lifestyle and robust immune system may feel and function more like someone in their 40s, while another person of the same age with chronic health conditions might experience aging more rapidly. This highlights that the point at which one "gets old" is not a fixed biological threshold but a dynamic interplay of genetic, environmental, and behavioral factors.

Psychologically, the concept of aging is heavily influenced by attitudes, beliefs, and self-perception. The phrase "I feel old" is a common expression that reflects a subjective experience rather than a fact. For instance, someone who has retired and feels a sense of loss or regret about missed opportunities might perceive themselves as older due to emotional and psychological factors, even if their physical health is age-appropriate. Conversely, individuals who maintain a sense of purpose, engage in lifelong learning, and have strong social connections often report feeling younger, as their mindset and quality of life mitigate the effects of chronological age. This psychological aspect is crucial, as it shows that aging is not a one-size-fits-all process but a highly personalized journey shaped by one's inner state.

Sociocultural factors also play a significant role in defining when someone is considered "old." In many societies, there are cultural stereotypes and ageist attitudes that associate old age with decline, dependency, or loss of value. For example, the retirement age in different countries varies, with some cultures valuing work into later life, while others see retirement as a natural part of aging. These cultural norms can influence how individuals perceive their own age and the age of others. For instance, in some Indigenous cultures, older individuals

are highly respected for their wisdom and experience, whereas in Western cultures, there is often a stigma around aging, which can lead to feelings of inadequacy or marginalization.

The role of health and lifestyle is another critical factor. Chronic diseases like diabetes, heart disease, or Alzheimer's can accelerate the aging process, both physically and mentally, by limiting mobility, cognitive function, and emotional well-being.However, lifestyle choices such as a healthy diet, regular exercise, and stress management can delay these processes, keeping individuals vibrant and active well into their later years. For example, a 70-year-old who exercises daily and has a balanced diet might have the physical and mental agility of someone in their 50s, challenging the notion that age is determined by health alone.

In essence, the point at which a person gets "old" is a multifaceted interplay of biology, psychology, sociology, and culture. It is not a single, definitive moment but a continuous process that varies from person to person. Age is a social construct that is shaped by how society, culture, and the individual perceive and experience time. To answer this question comprehensively, one must consider the biological underpinnings of aging, the psychological and emotional factors that influence self-perception, the cultural norms that define age, and the lifestyle choices that either accelerate or mitigate the aging process. By understanding these dimensions, we can move beyond stereotypes and embrace aging as a natural, diverse, and personally meaningful journey, rather than a uniform state defined by a single age point.

**Rubrics:**
Part 1: Writing Quality (3 points total)

1.1 Clarity and Logical Flow (2 points)
Excellent (2 pts): The response is well-structured with smooth transitions between ideas. Example: "While biological aging is inevitable, the psychological experience of 'feeling old' often stems from a loss of purpose—such as retirement or social disengagement—rather than chronological age alone. This aligns with activity theory, which posits that maintaining meaningful roles delays the subjective sense of aging."
Needs Improvement (0–1 pt): The response lacks cohesion or jumps abruptly between points. Example: "Old age is about health. Some people regret things. Also, society treats you differently."

1.2 Engagement and Stylistic Flair (1 point)
Excellent (1 pt): Uses vivid language, analogies, or a compelling voice. Example: "Aging is less about the number of candles on a cake and more about the weight of unlived dreams—a 30-year-old with chronic illness may feel 'older' than a vibrant 70-year-old scaling mountains."
Needs Improvement (0 pts): Dry or overly generic. Example: "Old age is defined by physical decline and societal expectations."

Part 2: Content Coverage (7 points total)

2.1 Biological Factors (1.5 points)
Excellent (1.5 pts): Discusses specific health markers (e.g., telomere shortening, chronic illness) and their variability.
Sufficient (1 pt): Mentions health generally but lacks detail.
Missing (0 pts): Omits biological factors entirely.

2.2 Psychological and Emotional Dimensions (1.5 points)
Excellent (1.5 pts): Explores attitudes (e.g., Erikson's "integrity vs. despair"), regrets, or resilience.
Sufficient (1 pt): Briefly notes mindset but lacks analysis.
Missing (0 pts): No discussion of psychological aspects.

2.3 Sociocultural Influences (1 point)
Excellent (1 pt): Analyzes how culture and media shape aging perceptions (e.g., ageism, "young-old" distinctions).
Sufficient (0.5 pt): Mentions society superficially.

Missing (0 pts): Ignores sociocultural context.

2.4 Subjective vs. Chronological Age (1 point)
Excellent (1 pt): Contrasts lived experience with numerical age. Example: "A 50-year-old veteran with PTSD may feel older than a 75-year-old artist."
Sufficient (0.5 pt): Briefly acknowledges subjectivity.
Missing (0 pts): Treats aging as purely objective.

2.5 Theories of Aging (1 point)
Excellent (1 pt): References theories (e.g., activity theory, disengagement theory) with application.
Sufficient (0.5 pt): Names theories without elaboration.
Missing (0 pts): No theoretical framework.

2.6 Variability and Exceptions (1 point)
Excellent (1 pt): Highlights outliers (e.g., centenarians, early-onset "old" feelings due to trauma).
Sufficient (0.5 pt): Notes variability but lacks examples.
Missing (0 pts): Presents aging as uniform.

## Case 3

**Question:** In a story I am writing, I want a criminal organization to be the antagonist. What I'm planning is start it with the reader thinking the organization is good, but as they keep reading, they start to think that the organization is bad. Any advice?

**Answer Before Train:**

...
**Establish the organization's good reputation**: Begin by introducing the organization in a way that makes it seem like a positive force. Show how they help people, provide services, or work towards a noble cause. This will create a positive image in the reader's mind.

...
**Gradually reveal the dark side**: As the story progresses, start to reveal information that contradicts the organization's good reputation. This could be through a character's actions, dialogue, or internal thoughts. For example, you could show that the organization is involved in illegal activities, has a hidden agenda, or is corrupt.

...
**Reveal the twist**: At a crucial moment in the story, reveal the full extent of the organization's true nature. This could be a dramatic scene where the protagonist discovers the organization's dark secrets, or a character's confession that reveals the truth. This twist should be significant enough to change the reader's perception of the organization.

...
**Answer After Train:**

...
The story begins by introducing the organization as a seemingly benevolent entity, often positioned as a savior in a community plagued by poverty or social inequality. For instance, the group might start as a local charity, providing essential services like food, employment, or educational support. This initial phase is crucial for building the organization's credibility and emotional investment from the reader. The members are depicted as dedicated, compassionate individuals, perhaps including a protagonist who is deeply involved, rationalizing their actions as necessary for the greater good. This setup creates a moral ambiguity, as the organization's good deeds blur the lines between right and wrong, making the reader initially sympathize with their cause.

...
A turning point occurs when the group begins protecting illegal activities, such as drug trafficking or money laundering, under the guise of their charitable mission. This shift is not abrupt but a gradual erosion of their initial ideals. The members, including the protagonist, justify their actions with arguments about survival, economic necessity, or a perceived duty

to their community. These justifications are presented with a mix of rationality and emotional appeal, making the reader question their initial assumptions. The organization's leaders might even use propaganda or manipulative tactics to maintain control, further entrenching the members' loyalty and the reader's confusion.

...

The climax of the twist comes in a confrontation that reveals the full extent of the organization's crimes. This could be a scene where the protagonist is forced to choose between their loyalty to the group and their moral conscience, leading to a dramatic conflict. The revelation of the organization's true nature—its role in a larger criminal network, the harm it causes to the community, and the personal sacrifices of the members—should be both unexpected and logically connected to the preceding events. This moment not only shocks the reader but also provides a deeper understanding of the characters' motivations, highlighting the complexities of morality in a world of gray areas.

...

**Rubrics:**
Part 1: Writing Quality (3 points total)

1.1 Clarity and Logical Flow (2 points)
Excellent (2 pts): The answer transitions smoothly from setup (initial positive portrayal) to payoff (revealing the organization's true nature). Example: "Begin by showing the organization funding schools and hospitals, then subtly introduce rumors of missing dissenters—letting the reader connect the dots."
Needs Improvement (0 pts): Ideas are disjointed or lack cause-and-effect progression. Example: "The group seems nice. Later, they're bad. Use foreshadowing."

1.2 Engagement and Stylistic Flair (1 point)
Excellent (1 pt): Uses vivid language, analogies, or narrative techniques to hook the reader. Example: "Like a slow poison, let their charity mask their coercion—each act of kindness a thread in a tightening noose."
Needs Improvement (0 pts): Dry or overly generic advice. Example: "Make them do good things first, then bad things."

Part 2: Content Coverage (7 points total)

2.1 Initial Positive Facade (1 point)
Excellent (1 pt): Describes concrete ways to establish the organization's benevolent image (e.g., philanthropy, charismatic leaders). Example: "Show them rebuilding a town after a disaster, with media praising their 'selflessness.'"
Sufficient (0.5 pt): Mentions positive traits but lacks detail.
Missing (0 pts): Omits this aspect.

2.2 Gradual Reveal of Corruption (2 points)
Excellent (2 pts): Explains how to drip-feed clues (e.g., unreliable narrators, conflicting evidence). Example: "Have a protagonist uncover discrepancies in their finances, or a trusted member vanish after asking questions."
Sufficient (1 pt): Suggests a reveal but lacks nuance.
Missing (0 pts): No discussion of pacing or techniques.

2.3 Moral Complexity (1 point)
Excellent (1 pt): Addresses how to make the organization's shift believable (e.g., internal factions, justified extremism). Example: "Their leader might rationalize violence as 'for the greater good,' making their fall tragic."
Sufficient (0.5 pt): Briefly mentions motives without depth.
Missing (0 pts): Ignores moral layers.

2.4 Character Arcs (1 point)

Excellent (1 pt): Ties the reveal to protagonist growth (e.g., disillusionment, betrayal). Example: "The protagonist's mentor is exposed as an enforcer, forcing them to question loyalty."
Sufficient (0.5 pt): Notes character impact vaguely.
Missing (0 pts): No connection to characters.

2.5 Foreshadowing and Subtlety (1 point)
Excellent (1 pt): Recommends specific techniques (e.g., symbolic imagery, offhand remarks). Example: "Early on, a minor character jokes about 'owing favors'—later revealed as blackmail."
Sufficient (0.5 pt): Mentions foreshadowing without examples.
Missing (0 pts): Absent.

2.6 Contrast with True Antagonists (1 point)
Excellent (1 pt): Compares the organization to overt villains to highlight their hypocrisy. Example: "While gangsters terrorize the streets, the organization 'negotiates peace'—but their terms include silent obedience."
Sufficient (0.5 pt): Briefly contrasts without elaboration.
Missing (0 pts): No comparison.

## A.5 PROMPT TEMPLATE

**Extract Meta Description**

Please identify and extract up to 3-5 well-written, engaging, or insightful passages from the text below that are directly relevant to the topic: "{topic/}".

These passages should be examples of high-quality writing that could enhance a reader's understanding, capture their interest, or exemplify a good writing style (e.g., vivid language, clear explanations, compelling narrative).

Each passage should reflect how a human writer would approach the topic, including the various angles, aspects, and depth of thought they might consider. The passage should showcase a natural flow of ideas, the use of persuasive or descriptive techniques, and a perspective that adds value to the reader's understanding. Focus on extracting contiguous blocks of text that stand alone as good examples of writing and effectively convey a complete thought or argument in a way that mirrors human writing styles.

Avoid extracting very short, fragmented phrases or simple factual lists unless they are exceptionally well-phrased and illustrative of good writing while also capturing a human writer's perspective on the topic.

Return the output as a JSON array of strings, where each string is an extracted passage that is a complete narrative segment containing both description and view, written in a manner akin to how a human writer would present it.

For example: ["complete narrative segment 1 text...", "complete narrative segment 2 text..."]
Text: {chunkcontent/}

**Example Meta Descriptions:**
Open-Ended QA isn't just about delivering factual correctness — it's about weaving context, tone, and reader engagement into answers. A well-crafted answer acknowledges nuance, anticipates follow-up questions, and balances clarity with depth. The reader should feel as though they're conversing with someone who understands not just the 'what' but also the 'why', making the dialogue both informative and resonant.

Figure 5: Prompt used for extract meta description

---

**Rubric Consolidation Instruction**

You are given multiple rubric sets $R^{(1)}, R^{(2)}, \ldots, R^{(K)}$ generated for the same open-ended question $q$. Your task is to consolidate them into a single high-quality rubric set $R^*(q)$. Please follow these steps:

1. **Identify Consensus**: Prioritize rubrics that appear frequently across different sets or are semantically equivalent.

2. **Merge Similar Rubrics**: Combine rubrics with overlapping meaning into a single, clear formulation.

3. **Preserve Discriminative Power**: Retain rubrics that can differentiate between high- and low-quality responses.

4. **Ensure Completeness and Clarity**: The final rubric set should comprehensively cover both content quality and writing quality dimensions, while avoiding redundancy.

**Output Format:** Provide the final rubric set $R^*(q)$ as a structured list of rubrics, each written as an explicit and self-contained evaluation criterion.

---

**Reference for LLM as Judge without Rubrics**

You are an expert in evaluating text quality. Please evaluate the quality of an AI assistant's response to a user's question. Be as strict as possible.
You need to evaluate across the following six dimensions, with scores ranging from 0 to 2. The scoring criteria from 0 to 2 for each dimension are as follows:

1. Relevance: From content highly relevant and fully applicable to the user's request to completely irrelevant or inapplicable.
2. Accuracy: From content completely accurate with no factual errors or misleading information to content with numerous errors and highly misleading.
3. Coherence: From clear structure with smooth logical connections to disorganized structure with no coherence.
4. Clarity: From clear language, rich in detail, and easy to understand to confusing expression with minimal details.
5. Breadth and Depth: From both broad and deep content with a lot of information to seriously lacking breadth and depth with minimal information.
6. Reading Experience: From excellent reading experience, engaging and easy to understand content to very poor reading experience, boring and hard to understand content.

Please evaluate the quality of the following response to a question according to the above requirements.
Please evaluate the quality of the response. The output must strictly follow the JSON format: {{"Analysis": ..., "Relevance": ..., "Accuracy": ..., "Coherence": ..., "Clarity": ..., "Breadth and Depth": ..., "Reading Experience": ...}}.

Ensure that only one integer between 0 and 2 is output for each dimension score.
Question: {question}
Response to be Graded: {answer}

---

Figure 7: Prompt used for rubric consolidation.

