# OpenReview forum: "QuRL: Rubrics As Judge For Open-Ended Question Answering"
_ICLR.cc/2026/Conference — ICLR 2026 Poster_

### Official Review · Reviewer_Kkv7 · 2025-10-14

**Soundness:** 2
**Presentation:** 2
**Contribution:** 2
**Rating:** 6
**Confidence:** 2

**Summary:**

The authors propose a new/novel approach to improving the performance of LLMs on Open Ended QA. This paper proposes QuRL (Open-Ended Question Answering with Rubric-guided Reinforcement Learning), a framework that transforms human-authored web sources into case-wise rubrics.

The key contributions are:
• QuRL, a framework that leverages internet text to construct case-wise rubrics as reward signals for open-ended question answering
• With the assistance of QuRL, two new datasets have been created. QuRL-Train dataset consisting of 800 Question–Rubric pairs, along with a QuRL-Test dataset of 400 entries that underwent human verification.

**Strengths:**

The main strength of the paper is in proposing a novel method to obtain rubrics on a case basis and use these rubrics to help train the LLMs to give an answer that is more human like. The authors have conducted various experiments using a variety of LLMs comparing their approach to some existing benchmarks. Thorough implementation details and results are a notable strength. The results indicate that QuRL achieves strong correlation with human judgments, outperforming other existing approaches.

**Weaknesses:**

Nothing major however the impact of the work is difficult to understand from a practical usecase point of view. It looks like the authors have not studied the case where facts are misrepresented in an answer by the LLM and how the rubrics deal to penalize such behavior.

**Questions:**

1. The paper would become easier to follow if a couple of examples for rubrics generated are shown as part of the main paper.
2. A minor typo in line 90 - Deepseek

---

> ### Author Response · Authors · 2025-11-19
> **Thanks for your sincere feedback!**
>
> Thank you for your sincere review. We are delighted to learn that you find our proposed method novel, and that we have provided detailed implementation specifics along with credible experimental comparisons. We are pleased to address your concerns regarding the paper.
>
> ### Q1: It looks like the authors have not studied the case where facts are misrepresented in an answer by the LLM and how the rubrics deal to penalize such behavior.
>
> - Thank you for your concern. When a model presents hallucinated facts in its responses, it indeed severely compromises the quality of the answer, which is a fundamental aspect to consider in open-ended question-answering tasks. In our rubric design, we have incorporated considerations for content coverage. As shown in Figure 3, the "Content Coverage" criterion requires the model's response to address the key aspects of the specific question, which encourages the model to generate facts consistent with the retrieved online information during the RL process. Additionally, the reason we did not introduce explicit supervision signals for factual hallucinations in the RL process is that we observed the base model already demonstrated a high degree of factual consistency even before training.
> - We randomly selected 30 questions from QuRL-Test and evaluated the factual consistency in the responses generated by Qwen2.5-7B-Instruct. Our assessment method involved using Gemini-2.5-Pro's online retrieval tool to perform factual consistency checks. The experiment results are shown below.
>
> | Model Variant                         | Factual Accuracy | Avg Performance |
> | ------------------------------------- | ---------------- | --------------- |
> | Qwen2.5-7B-Instruct (coldstart)       | 98.60%           | 34.1            |
> | Qwen2.5-7B-QuRL (Standard)            | 97.70%           | 59.3            |
> | Qwen2.5-7B-QuRL (w/ Fact Supervision) | 98.20%           | 57.9            |
>
> The experiment results indicate that the original Qwen2.5-7B-Instruct (coldstart) exhibits a high factual accuary (98.6%).  Without supervision of factuality, `Qwen2.5-7B-QuRL (Standard)` only exhibits a negligible drop of factual accuracy (97.7%), but achieves the best average performance. We experimented adding explicit factual supervision (`w/ Fact Supervision`), which slightly improves accuracy to 98.2%, but results in a slight decrease in overall performance (57.9 vs. 59.3) and incurs a nearly **fourfold increase in training costs**. Therefore, we humbly conclude that incorporating explicit factual supervision offers diminishing returns and is not necessary for maintaining the practical effectiveness of our framework.
>
> ### Q2: The paper would become easier to follow if a couple of examples for rubrics generated are shown as part of the main paper.
>
> - Thank you for your suggestions on improving the paper. In Section 4.5, we provide a case study of rubrics and corresponding questions along with effectiveness analysis. Additional case analyses are also included in Appendix A.4. Although the rebuttal phase allows for one additional page of supplementary content, the complete case analysis requires substantial space, making it hard to include more case studies in the main text. While we are unable to provide complete additional examples in the main text, **we have utilized the additional space available to include the guiding principles for rubric generation (Figure 3) in the paper to enhance its readability**.
>
> ### Q3:  Minor typo in line 90 - Deepseek
>
> - We have fixed the typo in the new version of our submission.

---

> > ### Comment · Reviewer_Kkv7 · 2025-11-24
> >
> > Thank you for the response to my comments. I still feel my earlier evaluation is apt.

---

> > > ### Author Response · Authors · 2025-11-25
> > > **Thank you for your reply！**
> > >
> > > We are glad that we were able to address some of your concerns, and we warmly welcome further discussion at any time.

---

### Official Review · Reviewer_51nw · 2025-11-01

**Soundness:** 3
**Presentation:** 3
**Contribution:** 3
**Rating:** 8
**Confidence:** 4

**Summary:**

This paper proposes a rubrics guided RL based framework for open-ended QA. Authors claim that the GRPO based approach with case-wise rubrics significantly improves QA performance on multiple evaluation benchmarks. The main contributions of this paper are as follows:
1.QuRL, a framework which leverages internet text to build case-wise rubrics as reward signal for open-ended QA.
2. A new QuRL-Train dataset with question-rubrics pair, along with a human verified test set for evaluation

**Strengths:**

The main strengths of the paper are as follows:
1. The question-rubric train and test set might be useful for researchers.
2. Rubric based reward models seems to have some novelty and it is interesting.
3. Detailed experiments and comparison with SOTA LLMs to support the claim that QuRL improves the avg performance on multiple benchmarks
4. Human verification of the dataset improved trust and increases quality.
5. Consistency analysis between automatic evaluation method and human judgements is interesting.

**Weaknesses:**

The main weaknesses of the paper are as follows:
1. The proposed approach used search engine returned results for initial evidences.
2. LLMs for meta-description generation might have its own error and biases.
3. Designs of rubrics might be subjective and might have some generalizability issues.

**Questions:**

1. How rubric parameters are decided? is it domain specific?
2. How strong are the reward signal coming from the rubrics?
3. How does agentic QA methods like react compare against QuRL?

---

> ### Author Response · Authors · 2025-11-19
> **Thanks for your sincere feedback (1/2)!**
>
> Thank you for your careful review. We are delighted to learn that you find our method novel, the proposed dataset valuable to the community, and the conducted experiments compelling. Your recognition means a great deal to us, and we will now address your questions below.
>
> ### Q1: How are rubric parameters decided? is it domain specific?
>
> - Our rubric generation follows a question-specific generation process, not limited to a specific domain. We begin by obtaining relevant online materials for each specific question, then proceed with a unified operational workflow and rubric generation principles to synthesize the rubrics. If we understand correctly, your concern refers to the hyperparameter settings used for rubric collection. In our implementation, we maintained the same hyperparameter configuration for every question. The entire process does not involve any domain-specific elements. Should you have any additional confusion regarding some particular parameters, we would be happy to provide further clarification.
>
> ### Q2:How strong are the reward signal coming from the rubrics?
>
> - Thanks for your insightful question. In response to your question, we will elaborate on three key aspects that demonstrate why rubric-based reward signals outperform traditional reward signals, which also reflects their "strong" ability. From the perspective of numerical sensitivity, reward signals generated based on rubrics exhibit higher variance characteristics. We have statistically analyzed the average scoring variance of four responses generated by the Qwen2.5-72B-Instruct model under temperature=1.0 on the QURL-test dataset, using three evaluation methods: traditional reward models, LLM-as-a-Judge without rubrics, and LLM-as-a-Judge with rubrics. As shown in the table below, rubric-based reward signals exhibit relatively higher variance, which is beneficial for GRPO training. Previous research [1] has demonstrated that reward signals with greater discriminative power facilitate model convergence in GRPO.
>
> | Metric   | LLM (w/o Rubrics) | Reward Model (internlm2-7b-reward) | LLM (w/ QuRL Rubrics) |
> | -------- | ----------------- | ---------------------------------- | --------------------- |
> | Variance | 0.45              | 0.91                               | **1.14**                  |
>
> - Secondly, as analyzed in Section 4.3, the reward signals generated by our method exhibit stronger alignment with human preferences. This implies that our approach can guide the model toward optimization directions that better reflect human preferences during the training process.
> - Lastly, based on the two points above, we have also demonstrated in evaluation experiments (Table 1) that the reward signals derived from rubrics enable the model to achieve optimal post-training performance (compared to traditional reward models and variants without rubrics).
> We hope the above clarifications have addressed your concerns.
>
> ### Q3: How do agentic QA methods like react compare against QuRL?
>
> - Thank you for raising this important point. It is crucial to clarify that **a direct comparison between QuRL and existing agentic QA methods may not be entirely equitable**, given their fundamentally different task objectives.
> - Current agentic QA approaches (such as websailor [2]) are primarily designed for tasks with verifiable reference answers (e.g., multi-hop reasoning or information retrieval), where the focus lies in optimizing intermediate reasoning trajectories to reach a definitive correct conclusion. In contrast, QuRL mainly addresses the distinct challenge of Open-Ended QA, where no single "gold answer" exists. Rather than optimizing tool usage or retrieval steps, QuRL establishes quantifiable and verifiable reward signals by automatically generating case-specific rubrics, thereby enabling Reinforcement Learning from Verifiable Rewards (RLVR) to enhance open-ended generation quality. While our current framework differs in its core design from agentic approaches, we recognize the potential value of integrating agentic workflows into open-ended scenarios as a promising direction for future research and thank you for your suggestion.

---

> ### Author Response · Authors · 2025-11-20
> **Thanks for your sincere feedback (2/2)!**
>
> ### Q4: LLMs for meta-description generation might have its own error and biases and Designs of rubrics might be subjective and might have some generalizability issues.
>
> - Thank you for your concern. In Appendix Figure 5, we present the prompt used to request the model to extract meta-descriptions. The prompt explicitly instructs the model to extract the writing content of human authors (including perspectives and writing techniques). We have observed that current mainstream open-source models (such as Qwen2.5-7B) can faithfully accomplish this task. Additionally, in Line 1214, we provide an example of a meta-description to support our observation.
> - Regarding the issue of rubric synthesis, it is possible that the process may introduce biases inherent to the LLM itself. In Lines 250-255, we describe our approach to mitigating this effect: we generate multiple rubrics for each question and subsequently synthesize them through variance analysis and majority voting. As demonstrated in Table 1, this method leads to performance gains, indicating its effectiveness in rubric synthesis. We acknowledge that there remains ample room for exploration in addressing inherent LLM biases, and this will be a focus of our future work.
>
> **Reference**
>
> [1] Yu, Qiying, et al. "Dapo: An open-source llm reinforcement learning system at scale." *arXiv preprint arXiv:2503.14476* (2025).
>
> [2] Li, Kuan, et al. "WebSailor: Navigating Super-human Reasoning for Web Agent." *arXiv preprint arXiv:2507.02592* (2025).

---

### Official Review · Reviewer_6oTe · 2025-11-02

**Soundness:** 2
**Presentation:** 2
**Contribution:** 2
**Rating:** 2
**Confidence:** 4

**Summary:**

This paper examines the automatic creation of rubrics for a given open-ended question $q$.  The idea is to induce a set of useful rubrics to score generation output so as to allow a RLVR-like mechanism to apply for such questions.  The idea proxies search engine API ranking results as a means of implicitly creating preferential document listing that applies to $q$.   Finally with the stable of metrics derived from a limited training set (also provided by the authors), the authors apply GPRO as the RL to tune output, resulting in visible improvements to 3 benchmarks.

**Strengths:**

* Contributes a training set for RL tuning of such open-ended questions (albeit quite small: 800 train/400 test)
* Substantial baselining against a range of models for useful input.
* Conducts a human evaluation to help assess consistency, shows that the correlation is not high (although not discussed much).
* Contributes a useful introspection of the aligned / created metrics in the **4.5 Case Study** section.

**Weaknesses:**

* The submission is generally ok, but suffers from a lack of proofediting: the arguments are not crisp and precise.
* In the face of prior work (see **Questions**), I find the work incremental, as there have been directly relevant work on inferring rubrics from question.
* The work implies that the metrics are done _per question_ and as described, over _multiple runs_ (but I may be mistaken, since the training seems to be done over the entire set of rubrics for the training set.  This seems overly expensive, but the inference cost of creating such metrics is not discussed.
* Some main text figures reference key metadata only present in the Appendices.  This makes the paper reliant on the supplemental materials and hence I would judge it not fitting in the length requirements for ICLR.

Other minor problems
* The "contributions" final paragraph of the intro overlaps significantly with the rest of the work, it'd be better to leave it out or at least de-duplicate the text with the rest of the intro.
* 090: Deepseek is misspelled.
* 121-124: your claim should mention on what data.  The gains could not be properly scoped otherwise.
* It seems you use $\LaTeX$ for typesetting.  Please use the appropriate quote structure (e.g., 301-303 caption)
* The ethics statement is mis-used.  The ethics block should not be used to describe annotation guidelines and data capture; that should be part of the normal space requirements.
* References should be appropriately checked for correct formatting.  Many titles have capitalisation protection errors (e.g. "llms" vs. "LLMs").

**Questions:**

* There are a few relevant works that work on inferring metrics from tasks that don't have specific metrics, which are relevant means to induce guidelines/metrics that seem to be missing from related work.  See:

  * Minzhi Li, Zhengyuan Liu, Shumin Deng, Shafiq Joty, Nancy Chen, and Min-Yen Kan (2025) DnA-Eval: Enhancing Large Language Model Evaluation through Decomposition and Aggregation. In Proceedings of the 31st International Conference on Computational Linguistics, pages 2277–2290, Abu Dhabi, UAE. Association for Computational Linguistics.
  * Do Xuan Long, Duong Ngoc Yen, Do Xuan Trong, Luu Anh Tuan, Kenji Kawaguchi, Shafiq Joty, Min-Yen Kan and Nancy F. Chen (2025) Beyond In-Context Learning: Aligning Long-form Generation of Large Language Models via Task-Inherent Attribute Guidelines. In Proceedings of the Findings of the Association for Computational Linguistics (ACL '25), Vienna, Austria.

* 330: how important is the cold-start SFT to the work?  While not a contribution it is important to know.
* 296: is this meant to say `w/o rlfh`?  Also: I'm not sure that these are additive ablations or whether they are single ablations.  Notation might help.
*

---

> ### Author Response · Authors · 2025-11-19
> **Thanks for your valuable feedback! (1/2)**
>
> We are grateful for your thorough review! It's our privilege to learn that you consider our work as contributing a dataset beneficial to the field, and that it includes systematic evaluations of model performance and human relevance, along with compelling case studies. Regarding all your questions and concerns, we have carefully replied as follows.
>
> ### Q1: The paper lacks discussion of two relevant works that utilize rubrics or guidelines and it seems to be an incremental advancement of these methods.
>
> - Thank you for your concerns. We have carefully reviewed the relevant works by Li et al. (DnA-Eval) and Long et al. (LongGuide), both of which propose rubric generation methods, and we will cite them accordingly. While we acknowledge these contributions, we believe that our work, QuRL, demonstrates clear distinctions and represents more than an incremental advancement over the existing literature from the following aspects:
> - **Fundamental difference in the method**
>   - DnA-Eval requires generating predefined scoring rubrics for each task, then conducting dimensional scoring for two different responses to a single question, and finally aggregating the scores by weighting the dimensional scores according to the question. LongGuide requires summarizing two types of guidelines from a small training set to guide the scoring and answer format dimensions respectively, which are then used as rubrics for each question.
>   - Unlike the two aforementioned approaches, QuRL maintains rubric generation at the question level rather than attempting to identify universal rubrics for an entire category of tasks (Section 3.1). Moreover, it incorporates external information by leveraging corpora inherently present in human online resources for rubric generation. In contrast, the rubrics in the other two methods are derived from reference-free black-box LLM generation and this internal loop of LLMs can hardly yield substantial new information gain.
> - **A specialized framework for RL and open-ended challenges:**
>   - While both DnA-Eval and LongGuide offer "question-to-rubrics" frameworks, their designs present certain limitations in our target setting. DnA-Eval's reliance on pairwise comparisons makes it unsuitable for generating robust reward signals in GRPO training. Meanwhile, LongGuide—designed for in-context learning with reference-based tasks like summarization—lacks the ability to produce numerical scores for RL applications. Its structural constraints on response length also limit its applicability in open-ended creative scenarios where no gold-standard answers exist. In contrast, QuRL can generate scores that are more aligned with human ratings through rubrics (Table 2), and it has been demonstrated that such scores can effectively improve model performance in open-ended generation scenarios via reinforcement learning.
> - **More comprehensive analysis and reproducibility：**
>   - Compared to these two studies, our paper provides a more comprehensive discussion on the mapping approach from questions to rubrics. Regarding the rubric generation methodology, we conducted multiple ablation studies (Table 1) and exploratory discussions on rubric scoring mechanisms (Section 4.3). We believe these analyses offer deeper insights into the effectiveness of our approach.
>   - It is worth noting that neither of these two related works has provided open-source code for reproducibility. In contrast, we have provided an anonymous code repository, which we believe will be beneficial to the community.
>
> We are sorry for missing the two related studies and **have already included both works in the related work section of our new version paper**.
>
>
> ### Q2: Creating a dedicated rubric set for each question appears to be prohibitively expensive, and the original text lacks relevant discussion on cost implications.
>
> - Thanks for your question. In the Methodology section of our paper, we have provided a detailed description of the rubric generation process, including the models and hyperparameters selected for each stage, which we hope will facilitate your understanding. We also recognize the importance of discussing generation costs and will address the average cost of rubric generation per question in our subsequent analysis.
> - Our end-to-end cost for generating rubrics per question is approximately \\$0.56, which includes query retrieval (\\$0.03), meta-description extraction (\\$0.03), generating preliminary rubrics (\\$0.30), and calculating answer variance (\\$0.20). Factoring in the discard rate, producing our 800-sample training set incurs a total cost of approximately \\$650. This is considerably more cost-effective than traditional reward model training, which typically requires tens of thousands of human-annotated preference pairs. At a market rate of \\$0.15 per pair, the annotation cost of traditional reward model alone would exceed \\$4,000, not including the substantial computational resources for training.

---

> > ### Comment · Reviewer_6oTe · 2025-11-23
> > **Thank you for your rebuttal**
> >
> > Hi authors, I agree with your changes and clarifications.  I have updated my scores as a result.

---

> > > ### Author Response · Authors · 2025-11-25
> > > **Thank you for your reply.**
> > >
> > > We are pleased to know that our response has addressed your concerns, and we thank you again for your suggestions to improve the quality of our paper.

---

> ### Author Response · Authors · 2025-11-19
> **Thanks for your valuable feedback! (2/2)**
>
> ### Q3: Some parts of this paper reliant on the supplemental materials and hence I would judge it not fitting in the length requirements for ICLR ###
> - We apologize for the reviewing inconvenience and burden brought to you, but have to argue that our initial submission fully complies with the guidelines, as we find that previous ICLR submissions, including HelloBench [1] and LongWriter [2], also include framework prompts or complex processing pipelines in the appendix.  Due to page limits,  we have to put a lot of details in the appendix, and put the correponding citation in the main text for readibility. It appears to be a common practice for contemporary LLM conference papers, particularly as studies increasingly rely on elaborate prompt designs and multi-stage processing workflows. We believe the main text has succinctly presented our core contributions, while the appendix provides supplementary implementation details for the community.  We would sincerely appreciate your suggestion and agree that incorporating some text figures into the main body would enhance the paper. In the camera-ready version, we will have one additional page. Following your suggestion, **we have moved the rubrics design principles prompt (Line 190-208) from the appendix into the main text** to improve the readability of the paper. We hope this clarification could alleviate your concerns.
>
> ### Q4:How important is the cold-start SFT to the work? While not a contribution it is important to know. ###
> - As described in Section 4.1(Implementation Details), our cold-start step is designed to familiarize the model with the \<think\>\</think\> \<answer\>\</answer\> output format, facilitating efficient sampling during the RL training phase. Since only 64 training examples were used, this step does not significantly enhance the model's inherent capabilities. As shown in Table 1, the cold-start baseline achieved an average score of only 34.1, whereas after RL training, our model reached 59.3. Thus, the cold-start phase can be viewed as a lightweight fine-tuning process aimed primarily at format compliance.
>
> ### Q5: Is this meant to say w/o rlfh? Also: I'm not sure that these are additive ablations or whether they are single ablations.  ###
> - Thank you for your careful reading. We acknowledge that our experimental notation was not clearly explained. We conducted ablation studies in a singular manner, meaning that each variant listed was derived from the Qwen2.5-7B-QuRL baseline by applying exactly one modification at a time. "w/" is actually an abbreviation for "with". This variant indicates that the reward signals are derived from the reward model rather than from rubrics. Following your suggestion, **we have provided an explanation in the caption(Line 346-351)**.
>
> ### Q6:  A human evaluation to help assess consistency, shows that the correlation is not high? ###
> - Thanks for your concern. We would like to clarify that the Spearman correlation coefficient of 0.31 achieved by our method in Table 2 should not be considered low. In previous LLM-as-a-Judge studies [3][4], the reported Spearman correlation coefficients for tasks such as summarization typically range between 0.1 and 0.4. In open-ended question answering evaluations, the responses to be assessed are generally longer, making it more challenging for automated evaluation methods to align with human judgments. In Table 2, we compared the correlations of multiple methods, and QuRL demonstrated superior performance. We share your view that there is still room for improvement in aligning automated evaluation methods with human judgments, and we aim to further explore this in our future work.
>
> ### Q7: Other minor problems ###
> - Thank you sincerely for the detailed comments. **We have gone through each of the minor issues you raised and have now incorporated all corresponding corrections in the updated version of the manuscript**.
>
> **Reference**
>
> [1] Que, Haoran, et al. "Hellobench: Evaluating long text generation capabilities of large language models." arXiv preprint arXiv:2409.16191 (2024).
>
> [2] Bai, Yushi, et al. "Longwriter: Unleashing 10,000+ word generation from long context llms." arXiv preprint arXiv:2408.07055 (2024).
>
> [3] Jinlan Fu, See-Kiong Ng, Zhengbao Jiang, and Pengfei Liu. Gptscore: Evaluate as you desire. arXiv preprint arXiv:2302.04166, 2023.
>
> [4] Raina, Vyas, Adian Liusie, and Mark Gales. "Is llm-as-a-judge robust? investigating universal adversarial attacks on zero-shot llm assessment." arXiv preprint arXiv:2402.14016 (2024).

---

### Author Response · Authors · 2025-12-01
**Summary of Paper Revision**

We sincerely appreciate all reviewers for their thorough evaluation and valuable suggestions. According to the suggestions, we have summarized the revisions made to the paper during the discussion phase as follows:

---
**Reviewer 6oTe**
1. We have added a figure illustrating the rubrics generation principles in Section 3 to enhance the readability (Line 191-210).
2. We have revised the caption of Table 1 to make the experimental setup clearer (Line 347-352).
3. We have revised the description of the contributions in the introduction (Line 114-124) and corrected typos and formatting errors throughout the article (Line 90).
4. We have added discussions on two new relevant works in the related work section (Line 157-158).

---
**Reviewer 51nw**

4. We have revised the narrative in Section 4.3 to explain why our reward signal performs better in the experiments (Line 431-434).

---
**Reviewer Kkv7**

5. We have revised Figure 3 and moved it from the supplementary materials to the main text to enhance readers' understanding of our method (Line 210).

---

### Author Response · Authors · 2025-12-02
**General Response！**

Dear AC, PC and Reviewers:

We sincerely thank you for your time and expertise, and we are grateful to all reviewers for their valuable feedback. We are pleased to learn that the reviewers consider we have conducted systematic experiments and case studies (**reviewer 6oTe 51nw Kkv7**); that the proposed method is novel and useful (**reviewer 51nw Kkv7**); and that the proposed approach can provide a helpful dataset for the community (**reviewer 51nw Kkv7**).

During the discussion period, we addressed each of the reviewers' comments and concerns point by point. We summarize the main clarifications and revisions as follows:

- **Added the figure in the main text to aid in understanding the method:** Due to the page limit of the submission version, and to ensure the completeness of the paper, we had to place some of the prompt templates used in the process in the supplementary material. Since the discussion phase allowed for expanded content, we have incorporated the guiding principles for rubric synthesis into the paper (figure 3) as suggested by the reviewers, to facilitate readers' understanding of our overall pipeline.

- **Added discussions on two related works:** Reviewer 6oTe pointed out two works related to our method. After carefully reading and comparing them, we found that our work fundamentally differs from these two studies in terms of RL applicability, rubric generation methods, and analytical comprehensiveness (detailed in the discussion with Reviewer 6oTe). We have added discussions on these two methods in the related work section (Line 157) and consequently **received improved scores and recognition from the reviewer.**

- **Refinement of argument presentation:** We have systematically addressed the reviewers' suggestions regarding details in the paper, optimizing unclear expressions and correcting typos as detailed in the summary of revisions below.

Although reviewer 6oTe initially gave a low score, we think that this may be due to a misunderstanding of our original intention regarding article formatting and the distinctions between our work and related studies. After revising the article with expanded content and providing point-by-point clarifications, we are pleased to see that **reviewer 6oTe has raised overall ratings along with soundness and contribution scores (on 23 Nov, 11:28)**, and have acknowledged our article (details in the discussion below). At the same time, the two remaining **reviewers (51nw and Kkv7) who initially recommended our paper maintained their high evaluation of our article after the discussion**. We are deeply grateful for their recognition.

We fully understand that this is a particularly busy period for ACs. We hope these actual outcomes of the rebuttal process will be fully considered in your final decision-making. Deeply thank you again for your time and for handling this challenging situation.

Best regards,

The Authors

---

### Meta-Review · Area_Chair_Pqp5 · 2026-01-09

**Summary:**

This paper proposes a rubrics guided RL based framework for open-ended QA and shows that GRPO based approach with case-wise rubrics significantly improves QA performance. Reviewers agree that the proposed framework is novel and the contribution of the paper is valuable. The main concerns are
1. Creating a dedicated rubric set for each question appears to be prohibitively expensive.
2. When a model presents hallucinated facts in its responses, it indeed severely compromises the quality of the answer.

The overall quality of the paper is solid and the proposed techniques should be interesting to the community.

**Reviewer Concerns:**

The main concerns from the reviewers are mostly addressed in author response.

**Reviewer Scores:**

2,8,6

The reviewer with 2 is likely to increase the score after rebuttal.

---

### Decision · Program_Chairs · 2026-01-26

Accept (Poster)